# Speech Recognition and Meaning Interpretation: Towards Disambiguation of Structurally Ambiguous Spoken Utterances in Indonesian

**Ruhiyah Faradishi Widiaputri[1]***, **Ayu Purwarianti[1], Dessi Puji Lestari[1],**
**Kurniawati Azizah[2], Dipta Tanaya[2], Sakriani Sakti[3]**

[1]Bandung Institute of Technology, Indonesia
[2]University of Indonesia, Indonesia
[3]Japan Advanced Institute of Science and Technology, Japan
`23523014@std.stei.itb.ac.id`, `{ayu,dessipuji}@itb.ac.id`,
`{kurniawati.azizah,diptatanaya}@cs.ui.ac.id`, `ssakti@jaist.ac.jp`

## Abstract

Despite being the world's fourth-most populous country, the development of spoken language technologies in Indonesia still needs improvement. Most automatic speech recognition (ASR) systems that have been developed are still limited to transcribing the exact word-by-word, which, in many cases, consists of ambiguous sentences. In fact, speakers use prosodic characteristics of speech to convey different interpretations, which, unfortunately, these systems often ignore. In this study, we attempt to resolve structurally ambiguous utterances into unambiguous texts in Indonesian using prosodic information. To the best of our knowledge, this might be the first study to address this problem in the ASR context. Our contributions include (1) collecting the Indonesian speech corpus on structurally ambiguous sentences[1]; (2) conducting a survey on how people disambiguate structurally ambiguous sentences presented in both text and speech forms; and (3) constructing an Indonesian ASR and meaning interpretation system by utilizing both cascade and direct approaches to map speech to text, along with two additional prosodic information signals (pause and pitch). The experimental results reveal that it is possible to disambiguate these utterances. In this study, the proposed cascade system, utilizing Mel-spectrograms concatenated with F0 and energy as input, achieved a disambiguation accuracy of 79.6%, while the proposed direct system with the same input yielded an even more impressive disambiguation accuracy of 82.2%.

## 1 Introduction

Ambiguity is one of the challenges in natural language processing. It has been observed that nearly

---

*This work was conducted while the first author was doing internship at HA3CI Laboratory, JAIST, Japan under JST Sakura Science Program.

[1]Our corpus is available at `https://github.com/ha3ci-lab/struct_amb_ind`

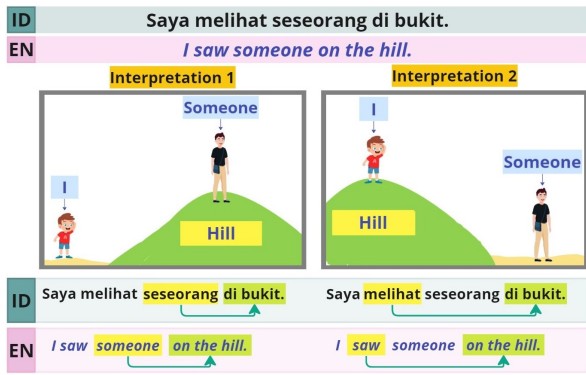

Figure 1: An example of a structurally ambiguous sentence in Indonesian and English.

every utterance contains some degree of ambiguity, even though alternate interpretations may not always be obvious to native speakers (Russell and Norvig, 2009). Hurford et al. (2007) categorized ambiguity into two main types: lexical ambiguity and structural ambiguity. Lexical ambiguity arises when a word has multiple meanings, as seen in the word "bank," which can refer to a financial institution or the side of a river. On the other hand, structural or syntactic ambiguity occurs when a phrase or sentence can be parsed in more than one way. For instance, in the sentence "I saw someone on the hill," the prepositional phrase "on the hill" can modify either (1) the verb "saw" or (2) the noun "someone." This structural ambiguity gives rise to semantic ambiguity, resulting in different possible meanings. An example of this is illustrated in Figure 1.

Efforts have been made to address the issue of ambiguity in natural language. However, the resolution of lexical ambiguity is more commonly studied than structural ambiguity. There is even a specific task dedicated to its resolution, namely word-sense disambiguation. Furthermore, most studies on the resolution of structural ambiguity are

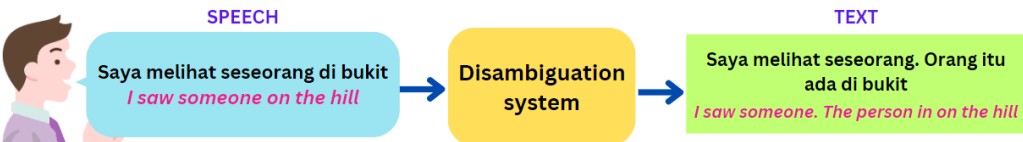

Figure 2: The disambiguation system.

focused on parsing tasks (Shieber, 1983; Li et al., 2014; Tran et al., 2018). To the best of our knowledge, there have been no studies directly yielding disambiguated sentences as output in the field of structural ambiguity resolution.

Speech is the most natural form of human-to-human communication. It is also considered the richest form of human communication in terms of bandwidth, conveying valuable supplementary information not found in text, particularly the prosodic structure (Tran et al., 2018), which listeners can use to resolve structural ambiguities (Price et al., 1991). Therefore, understanding spoken language is one of the earliest goals of natural language processing (Jurafsky and Martin, 2022). Today, automatic speech recognition (ASR) technology has advanced and is widely used. Unfortunately, most ASR systems do not typically address issues of ambiguity.

Consequently, in this work, we have made the first attempt to create a system capable of disambiguating structurally ambiguous utterances into unambiguous text by using the provided prosodic information. We propose to build the system by adapting both the cascade and direct approaches to speech-to-text mapping. The proposed cascade system combines two components: ASR and a novel text disambiguation (TD) model, while the proposed direct system utilizes another novel speech disambiguation (SD) model. The TD model is specifically designed to transform the ASR output, which is the transcription of structurally ambiguous utterances, into the intended unambiguous text. An illustration of the disambiguation system is shown in Figure 2.

Therefore, our contributions include: (1) constructing the first Indonesian structural ambiguity corpus; (2) proposing both cascade and direct approaches for speech-to-text mapping to build the disambiguation system; (3) performing experiments by developing the disambiguation systems using the Indonesian structural ambiguity corpus created with four proposed audio input combinations: Mel-spectrogram, Mel-spectrogram concatenated with F0, Mel-spectrogram concatenated with energy, and Mel-spectrogram concatenated with F0 and energy; (4) conducting human assessments to examine how individuals resolve structurally ambiguous sentences in written and spoken formats.

## 2 Related Work

Indonesia has one of the largest populations in the world (Cahyawijaya et al., 2021), yet the availability of resources and the progress of NLP research in Indonesian lag behind (Wilie et al., 2020; Cahyawijaya et al., 2022). Nevertheless, research on Indonesian, both speech and text, is gradually progressing despite the challenges mentioned earlier. Even though the majority of research and existing data in Indonesian comes from the text modality, several works have attempted some speech processing tasks in Indonesian. To date, some speech corpora have also been developed (Sakti et al., 2004; Lestari et al., 2006; Sakti et al., 2008a, 2013), as well as the development of ASR (Sakti et al., 2004, 2013; Ferdiansyah and Purwarianti, 2011; Prakoso et al., 2016; Cahyawijaya et al., 2022) and TTS (Sakti et al., 2008b; Azis et al., 2011; Mengko and Ayuningtyas, 2013; Azizah et al., 2020). Most Indonesian ASRs and TTSs have achieved good results. However, specifically for ASR tasks, most ASR that has been developed in Indonesian and also in other languages is still limited to transcribing the exact word-by-word, ignoring ambiguities in the transcription.

A number of attempts have also been made to address the issue of ambiguity in Indonesian, particularly in the text domain. With regard to word-sense disambiguation, several studies have been conducted (Uliniansyah and Ishizaki, 2006; Faisal et al., 2018; Mahendra et al., 2018). Nevertheless, research on structural ambiguity resolution, especially in speech, is far less common, and there is a notable gap in this area.

On the other hand, some studies have attempted to utilize prosodic information in speech to resolve ambiguity across various tasks. For example, Tran et al. (2018) developed a dependency parser that

| Type | Ambiguous sentences | Disambiguation sentences |
|---|---|---|
| 4 | The book on the chair I just bought is good. | (1) The book on the chair is good. I just bought the book. (2) The book on the chair is good. I just bought the chair. |
| 5 | They buried the treasure they found in the park. | (1) They buried the treasure they found. They buried it in the park. (2) They buried the treasure they found. They found it in the park. |
| 6 | I saw someone on the hill. | (1) I saw someone. I saw on the hill. (2) I saw someone. The person is on the hill. |
| 10 | I eat spicy chicken and eggs. | (1) I eat chicken and eggs. The chicken is spicy. (2) I eat chicken and eggs. The chicken and eggs are both spicy. |

Table 1: Examples of structurally ambiguous sentences and the results of their disambiguation for each adopted type.

incorporates acoustic-prosodic features, including pauses, duration, fundamental frequency (F0), and energy. Cho et al. (2019) attempted to classify structurally ambiguous sentences into seven categories of intentions using both acoustic and textual features. For the acoustic features the work combined root mean square energy (RMS energy) with Mel-spectrograms by concatenating them framewise. Tokuyama et al. (2021) addressed sentences with ambiguous intensity by adding an intensifier based on word emphasis. However, it's worth noting that most of these studies were conducted for major languages (e.g. English, Japanese, etc.). Furthermore, to the best of our knowledge, no studies have attempted to exploit prosodic information to disambiguate structurally ambiguous utterances that directly produce unambiguous texts. Therefore, we will address this issue in our study. Although our focus is on the Indonesian language, the approach can potentially be applied to other languages as well.

## 3 Proposed Method

### 3.1 Corpus construction

Because of the limited availability of documentation on structural ambiguity in the Indonesian language, we turned to English-language literature. Specifically, we drew upon the work of Taha (1983), who classified structural ambiguity in English into twelve types. However, we chose to adopt only four of these types, focusing on those most relevant to Indonesian linguistic characteristics:

1. **Type 4: noun + noun + modifier**. This type of sentence becomes ambiguous because the modifier can be considered to modify either the first noun or the second noun.

2. **Type 5: verb + verb + adverbial modifier**. Sentences of this type become ambiguous because the adverbial modifier can be considered to modify either the first or the second verb.

3. **Type 6: verb + noun + modifier**. In this case, the modifier can be interpreted as an adverb linked to the verb or as an adjective associated with the noun.

4. **Type 10: modifier + noun + conjunction + noun**. This type of sentence becomes ambiguous because the modifier can be applied to either the first noun or the second noun.

Through crowdsourcing, we created 100 structurally ambiguous sentences for each adopted type, with each ambiguous sentence having two possible interpretations. These interpretations were then transformed into unambiguous texts, consisting of two sentences each. The first sentence remained the same for interpretations derived from the same ambiguous sentence, while the second sentence differed to provide clarification for the intended interpretation. As a result, we obtained a total of 800 pairs of ambiguous sentences and unambiguous texts. Table 1 presents examples of ambiguous sentences and their corresponding unambiguous texts for each adopted type.

After validation by an Indonesian linguist, the 800 pairs of ambiguous sentences and their corresponding unambiguous texts were divided into 10 groups: 8 groups for training, 1 group for validation, and 1 group for the testing set. This division ensured that no sentences overlapped between the training, development, and test sets.

The utterances of 400 structurally ambiguous sentences were recorded in a controlled, low-noise office environment. The recording involved 22 speakers, consisting of 11 males and 11 females.

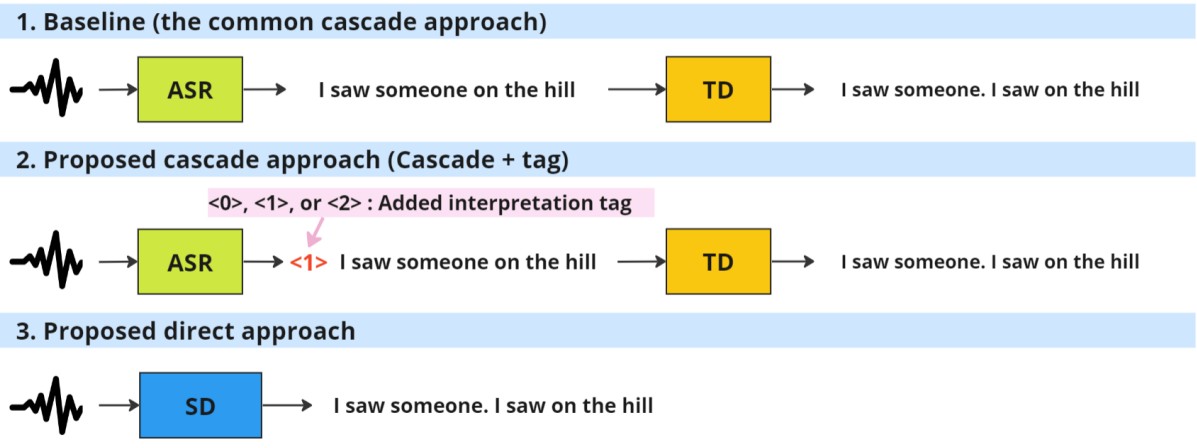

Figure 3: Proposed approaches for building the disambiguation system.

Each speaker was assigned to different groups of ambiguous sentences to ensure that neither speakers nor sentences overlapped across the training, development, and testing sets. For each pair of ambiguous sentences and their interpretations, the speaker was instructed to read the ambiguous sentence in a way that conveyed the intended interpretation naturally, without overemphasis. Before the recording session, we provided the speakers with the structurally ambiguous texts they would read, allowing them to study in advance. During the recording session, we closely monitored each speaker and requested corrections if their speech sounded unnatural due to overemphasis. The complete recording resulted in a total of 4,800 utterances.

### 3.2 Disambiguation of structurally ambiguous sentences in Indonesian

To develop a system capable of disambiguating structurally ambiguous utterances based on prosodic information, two key considerations arise: determining the necessary prosodic features for disambiguation and developing the disambiguation system.

#### 3.2.1 Utilization of prosodic information for disambiguation

According to Taha (1983), several prosodic signals that can be used to disambiguate structurally ambiguous sentences include emphasis, pitch, and juncture. Emphasis encompasses changes in pitch, duration, and energy during speech, as noted by Tokuyama et al. (2021). The term 'juncture' refers to speech features that allow listeners to detect word or phrase boundaries. Specifically, junctures that can be used to disambiguate structurally am-

biguous sentences are terminal junctures, characterized by a pitch change followed by a pause. Tran et al. (2018) mentioned four features that are widely used in computational prosody models: pauses, duration, fundamental frequency, and energy. Therefore, in this work, we focus specifically on two prosodic cues for disambiguating structurally ambiguous utterances: pauses and pitch variation. In addition to these features, we also extracted Mel-spectrogram for speech recognition.

The features extracted for pause in this work were Mel-spectrogram and energy. These features were chosen because they indirectly reflect the presence of pauses. In a low-noise environment, when a pause occurs, all channels in the Mel-spectrogram and energy exhibit values close to zero. To represent pitch, we extracted F0. These features were then concatenated with the Mel-spectrogram framewise, following a similar approach as described by Cho et al. (2019). Therefore, in this work, four input combinations were proposed: (1) Mel-spectrogram only; (2) Mel-spectrogram concatenated with F0; (3) Mel-spectrogram concatenated with energy (root mean square energy / RMS energy); and (4) Mel-spectrogram concatenated with F0 and energy (RMS energy).

#### 3.2.2 Development of disambiguation system

The disambiguation system functions as a speech-to-text mapping, taking structurally ambiguous utterances as input and producing unambiguous text as output. Similar to other speech-to-text mapping tasks, the disambiguation system can be implemented using two approaches: the cascade approach, which combines two or more components, and the direct approach, which transforms structurally ambiguous utterances into unambiguous

text without any intermediate representation. In this study, as we described earlier, we propose the cascade disambiguation system, which consists of two components: the automatic speech recognition (ASR) model and a novel model known as the text disambiguation (TD) model. Additionally, we introduce the direct disambiguation system, featuring another novel model called the speech disambiguation (SD) model.

It is evident that the disambiguation system approaches bear a resemblance to the approaches used in speech-to-text translation, which involves the automatic translation of utterances from one language to texts in another language (Bentivogli et al., 2021). The key difference lies in the replacement of the machine translation model (MT) with the text disambiguation model (TD). While the MT model was responsible for translating the transcription of ASR output, the TD model was specifically designed to convert the ASR output, consisting of structurally ambiguous sentences, into unambiguous text in the same language.

The proposed cascade approach may encounter challenges when standard ASR is applied without modification, leading to the loss of crucial prosodic information needed for disambiguating structurally ambiguous utterances. This prosodic information loss prevents TD from generating the desired unambiguous texts. Hence, to build the disambiguation system, modifications must be made so that the TD knows not only the structurally ambiguous transcription but also which interpretation is intended by the utterance based on the prosodic information given. In this work, we propose to do this by adding a special 'meaning tag' indicating the intended interpretation based on the utterance's prosodic information. Thus, in the proposed cascade approach, the ASR generates not only the transcription of structurally ambiguous utterances but also the special tags. Since each structurally ambiguous sentence in this work was limited to having exactly two interpretations, we proposed to add a 'meaning tag' for each sentence: $< 0 >$ for unambiguous sentences and $< 1 >$ or $< 2 >$ for structurally ambiguous sentences. The tagging as '1' or '2' must be uniform for each structurally ambiguous sentence of the same type:

1. **Type 4: noun + noun + modifier**: tag $< 1 >$ was given if the modifier was interpreted as modifying the first noun; tag $< 2 >$ was given if the modifier was interpreted as modifying the second noun.

2. **Type 5: verb + verb + adverbial modifier**: tag $< 1 >$ was given if the modifier was interpreted as modifying the first verb; tag $< 2 >$ was given if the modifier was interpreted as modifying the second verb.

3. **Type 6: verb + noun + modifier**: tag $< 1 >$ was given if the modifier was interpreted as modifying the verb; tag $< 2 >$ was given if the modifier was interpreted as modifying the noun.

4. **Type 10: modifier + noun + conjunction + noun**: tag $< 1 >$ was given if the modifier was interpreted as modifying only one noun; tag $< 2 >$ was given if the modifier was interpreted as modifying both nouns.

For comparison, we also developed the disambiguation system using the original cascade approach (with no tag addition) as a baseline. Regarding the proposed direct approach, which inherently preserves all the necessary information for generating unambiguous text, it required no significant modifications from the standard direct approach. Hence, the proposed architectures are illustrated in Figure 3.

To effectively train the ASR and SD models, relying solely on the structural ambiguity corpus we created is insufficient. These models require more data to recognize speech accurately. Therefore, in addition to using the structural ambiguity corpus, we incorporated another speech corpus for training both ASR and SD. The structural ambiguity corpus we created was also insufficient for training the TD model. To overcome this limitation, we employed the transfer learning technique, utilizing a pre-trained model for the TD. Specifically, we used BART (Lewis et al., 2020), a well-known pre-training model suitable for TD tasks due to its sequence-to-sequence architecture.

## 4 Experiment

### 4.1 Human assessment

We conducted a survey involving 20 participants to investigate how people disambiguate structural ambiguity in Indonesian. This investigation encompassed two conditions: one in which the sentences were presented as text only and another in which they were presented as speech. To evaluate how individuals disambiguate structurally ambiguous

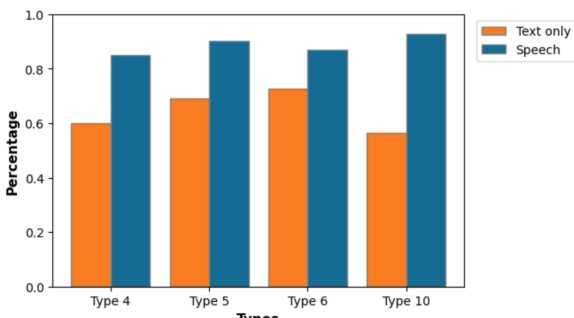

Figure 4: How unambiguous the sentences per type to humans in text and speech.

| | Indo-LVCSR | Type 4 | Type 5 | Type 6 | Type 10 | Total |
|---|---|---|---|---|---|---|
| Train | 41500 | 1040 | 1040 | 1040 | 1040 | 45660 |
| Dev | 160 | 80 | 80 | 80 | 80 | 480 |
| Test | 160 | 80 | 80 | 80 | 80 | 480 |

Table 2: Data used for training, development, and testing of ASR and SD models.

| | LVCSR (tag = 0) | Struct. amb tag = 1 | Struct. amb tag = 2 |
|---|---|---|---|
| Train | 320 | 80 x 4 types | 80 x 4 types |
| Dev | 40 | 10 x 4 types | 10 x 4 types |
| Test | 40 | 10 x 4 types | 10 x 4 types |

Table 3: Data used for training, development, and testing of TD models.

sentences in text form, each participant was presented with 40 ambiguous sentences and asked to select the most probable interpretation for each one. The disambiguation result for text-only sentences was determined by the interpretation chosen most frequently. To evaluate how people disambiguate structurally ambiguous sentences in speech form, each participant was asked to listen to 80 structurally ambiguous utterances and choose the most appropriate interpretation based on the recording. Participants were also given the opportunity to add their own interpretations if they believed they could provide a more precise representation than the two given options.

Figure 4 illustrates the degree of ambiguity perceived by the survey participants for each sentence type, both in text form and as speech. According to the survey results, structurally ambiguous sentences are difficult to disambiguate for people when presented in text form. The limited information conveyed solely through text often leads to subjective interpretations that vary among individuals. On the other hand, the majority of structurally ambiguous utterances can be correctly disambiguated by people when presented in speech. This suggests the presence of additional information in speech that may not be available in text alone, aiding individuals in disambiguating structurally ambiguous sentences. One source of this additional information is prosody.

### 4.2 Machine Evaluation
#### 4.2.1 Experimental set-up
All models utilized in this study were based on the Transformer architecture (Vaswani et al., 2017). We utilized the Speech-Transformer architecture (Dong et al., 2018) for both the ASR and SD models. The ASR component received audio input and generated transcriptions along with the meaning

tags, as mentioned earlier. The SD model also accepted audio input but produced unambiguous text as its output.

As mentioned previously, our training for both the ASR and SD models involved not only the structural ambiguity corpus we created but also an additional speech corpus. Specifically, we utilized the Indonesian LVCSR news corpus (Sakti et al., 2008a, 2004, 2013)[2]. This corpus consists of 400 speakers, each reading 110 sentences, resulting in more than 40 hours of speech. The sentences in the Indonesian LVCSR were unambiguous, so the additional meaning tag for the sentences was $< 0 >$, while the target output for the SD model for these sentences was the sentences themselves without any modifications. To ensure that there were no speakers or sentences that overlapped in the training, development, and test sets, we selected a subset of the LVCSR data. The specifics regarding the amount of data used for training, development, and testing of both the ASR and SD models can be found in Table 2.

In this work, we constructed the TD system by fine-tuning the IndoBART model (Cahyawijaya et al., 2021)[3]. The TD took the meaning tag outputted by the ASR concatenated with the ASR transcription, which is the structurally ambiguous sentence, as input and then outputted the unambiguous sentence.

The data used for the TD model consisted of 800 pairs of previously constructed structurally am-

[2]This corpus is publicly available at https://github.com/s-sakti/data_indsp_news_lvcsr

[3]https://github.com/IndoNLP/indonlg

|  | CER | WER | Tag acc. |
|---|---|---|---|
| **Mel** | | | |
| Baseline | 0.0621 | 0.2358 | - |
| ASR + tag | 0.0609 | 0.2140 | 0.9729 |
| **Mel + F0** | | | |
| Baseline | **0.0574** | 0.2157 | |
| ASR + tag | 0.0626 | 0.2294 | 0.975 |
| **Mel + RMS energy** | | | |
| Baseline | 0.0617 | 0.2247 | |
| ASR + tag | 0.0691 | 0.2375 | 0.9708 |
| **Mel + F0 + RMS energy** | | | |
| Baseline | 0.0638 | 0.2279 | |
| ASR + tag | 0.0575 | **0.2061** | **0.9875** |

Table 4: Comparison of CER, WER, and tag accuracy of the proposed ASR model with the baseline using four different input combinations.

|  |  | Baseline | TD + tag |
|---|---|---|---|
| | **BLEU** | 65.6 | **73.5** |
| | **ROUGE-1** | 91.7 | **96.3** |
| | **ROUGE-2** | 83.8 | **91.4** |
| | **ROUGE-L** | 91.7 | **96.2** |
| **Disam-biguati-on acc.** | Based on WER scores | 0.65 | **0.875** |
| | Based on BLEU scores | 0.658 | **0.908** |
| | Based on ROUGE-1 scores | 0.642 | **0.917** |
| | Based on ROUGE-2 scores | 0.642 | **0.883** |
| | Based on ROUGE-L scores | 0.642 | **0.925** |

Table 5: Comparison of BLEU, ROUGE, and disambiguation accuracy of the proposed TD model with the baseline.

biguous sentences, along with 400 unambiguous sentences taken from the transcription of the Indonesian LVCSR corpus. The development and test sets in TD shared the same sentences as ASR and SD, while the training set in TD only included a subset of sentences from the ASR and SD training data to ensure a balance of the meaning tags in the TD training data. Details about the amount of data used for TD training, validation, and testing can be found in Table 3.

The hyperparameter configurations used in this study are provided in detail in Appendix A.

#### 4.2.2 Experiment result

Table 4 displays the evaluation results for baseline ASR models and ASRs augmented with tags in four input combinations: (1) Mel-spectrogram; (2) Mel-spectrogram concatenated with F0; (3) Mel-spectrogram concatenated with RMS energy; and (4) Mel-spectrogram concatenated with F0 and RMS energy. The transcription produced by the ASRs was evaluated using CER (character error rate) and WER (word error rate), while the tags generated by the ASRs were assessed for accuracy. As shown in the table, most ASRs exhibited similar CERs and WERs, approximately 0.06 and 0.22, respectively, indicating satisfactory transcription performance. Similarly, the tagging accuracy showed a consistent trend, with accuracy surpassing 0.9, even for input Mel-spectrograms. This suggests that pauses, already captured in the Mel-spectrograms, already provide valuable information for disambiguating structurally ambiguous sentences. However, ASR with additional tags and

input Mel-spectrograms concatenated with F0 and RMS energy achieved slightly better WER and tag accuracy compared to other ASRs.

In this work, the evaluation of the TD models involved three types of metrics: BLEU, ROUGE, and the accuracy of the disambiguation results. BLEU (Papineni et al., 2002) and ROUGE (Lin, 2004) were selected due to the similarity of the TD task in this work to paraphrasing. The disambiguation results in this work were determined by classifying the unambiguous texts generated by the TD models based on their most suitable interpretation. The interpretation of an unambiguous text was determined using WER, BLEU, ROUGE-1, ROUGE-2, and ROUGE-L scores in comparison to three other texts: the non-disambiguated text, the disambiguated text with interpretation $< 1 >$, and the disambiguated text with interpretation $< 2 >$. This classification method is based on the intuition that the true TD output will exhibit the highest sentence similarity with the intended unambiguous text.

Table 5 presents the evaluation results of two TD models: the baseline TD with no tag addition and the TD with additional tags. The table clearly illustrates a significant improvement in TD performance achieved through the addition of tags, particularly in terms of disambiguation accuracy, which reaches approximately 90%. This under-

| | BLEU | ROUGE-1 | ROUGE-2 | ROUGE-L | Disambiguation accuracy | | | | |
| | | | | | Based on WER scores | Based on BLEU scores | Based on ROUGE-1 scores | Based on ROUGE-2 scores | Based on ROUGE-L scores |
|---|---|---|---|---|---|---|---|---|---|
| **Mel** | | | | | | | | | |
| Cascade baseline | 33.7 | 71.0 | 52.0 | 70.3 | 0.571 | 0.56 | 0.569 | 0.558 | 0.556 |
| Cascade + tag | **42.8** | 76.1 | 59.0 | 75.9 | 0.785 | 0.763 | 0.815 | 0.777 | 0.817 |
| Direct | 41.3 | 74.8 | 57.2 | 74.2 | 0.802 | **0.783** | 0.844 | 0.806 | 0.842 |
| **Mel + F0** | | | | | | | | | |
| Cascade baseline | 36.9 | 72.7 | 53.5 | 72.2 | 0.588 | 0.575 | 0.573 | 0.575 | 0.573 |
| Cascade + tag | 40.8 | 74.8 | 56.7 | 74.7 | 0.773 | 0.733 | 0.817 | 0.773 | 0.815 |
| Direct | 39.6 | 74.3 | 56.1 | 73.4 | 0.788 | 0.765 | 0.829 | 0.769 | 0.802 |
| **Mel + RMS energy** | | | | | | | | | |
| Cascade baseline | 35.2 | 71.7 | 53.1 | 71.2 | 0.59 | 0.592 | 0.583 | 0.585 | 0.579 |
| Cascade + tag | 39.4 | 73.7 | 55.7 | 73.5 | 0.781 | 0.742 | 0.817 | 0.771 | 0.815 |
| Direct | 41.2 | 74.4 | 56.6 | 73.5 | 0.821 | 0.765 | 0.84 | 0.8 | 0.827 |
| **Mel + F0 + RMS energy** | | | | | | | | | |
| Cascade baseline | 34.7 | 71.7 | 51.8 | 71.4 | 0.61 | 0.588 | 0.598 | 0.588 | 0.598 |
| **Cascade + tag** | **41.7** | **76.8** | **60.0** | **76.6** | **0.798** | **0.763** | **0.827** | **0.798** | **0.827** |
| **Direct** | **42.0** | **75.8** | **58.4** | **75.0** | **0.825** | **0.769** | **0.856** | **0.817** | **0.844** |

Table 6: Comparison of BLEU, ROUGE, and the disambiguation accuracy between the proposed cascade and direct models.

scores the crucial role of prosodic information, preserved by the tags, in enabling the TD model to generate the intended unambiguous text. The TD model achieves commendable scores, especially in ROUGE-1, ROUGE-2, and ROUGE-L, highlighting the positive impact of the transfer learning method. These high scores indicate a close alignment between the generated disambiguation text and the desired unambiguous text.

The cascade and direct systems were evaluated with the same metrics as TD: BLEU, ROUGE, and disambiguation accuracy. Table 6 shows the comparison between the cascade and direct systems built with four combinations of input. As can be seen in the table, both cascade and direct systems with the best performance use mel spectrograms combined with F0 and RMS energy as input, as in the case of ASR. Then, similar to the case of TD, the cascade systems with additional tags exhibit significantly better performance than the baseline cascade systems that do not include any additional tags.

As can be seen in Table 6, cascade models slightly outperform direct systems in terms of BLEU, ROUGE-1, ROUGE-2, and ROUGE-L scores. This makes sense considering that in the cascade approach, SD, which has a more challenging task, is trained using the same model and data as ASR without any transfer learning methods, while in the cascade approach, the task is divided between ASR and TD, and the TD models were pre-trained beforehand. However, the difference between both systems is not significant. This is because the output text from the disambiguated results is not considerably different from the ambiguous transcriptions, making it not very hard to learn. On the other hand, in terms of the accuracy of the disambiguation results, the direct systems demonstrate better performance, showing a difference of approximately 2.5%. This is due to the error propagation within the cascade system. The errors generated by the ASR, including transcription and tagging errors, result in a decrease in disambiguation accuracy of approximately 10% compared to the scenario where TD receives error-free transcription and tagging, as shown in Table 5.

In summary, in terms of average BLEU, ROUGE-1, ROUGE-2, and ROUGE-L scores, the proposed cascade system with Mel-spectrograms concatenated with F0 and energy (RMS energy) as input achieves a slightly higher score of 63.8. On the other hand, the proposed direct approach system with Mel-spectrograms concatenated with

F0 and RMS energy as input significantly improves disambiguation accuracy to 82.2%. Since our primary focus in this study is disambiguation, we can conclude that the direct system is the best in this case.

## 5 Conclusions

This paper introduces the first Indonesian structural ambiguity corpus, comprising 400 structurally ambiguous sentences, each paired with two unambiguous texts, totaling 4800 speech utterances. We also conducted an evaluation of human performance in disambiguating written and spoken structurally ambiguous sentences, highlighting the advantages of the spoken format. Additionally, we developed systems for disambiguating structurally ambiguous sentences in Indonesian by adapting both cascade and direct approaches to speech-to-text mapping, using two additional prosodic information signals (pauses and pitch). The experiment demonstrates that by utilizing prosodic information, structurally ambiguous sentences can be disambiguated into unambiguous interpretations. Among the systems tested, the proposed cascade system achieved average BLEU, ROUGE-1, ROUGE-2, and ROUGE-L scores of 63.8 and a disambiguation accuracy of 79.6%. In contrast, the proposed direct system, which uses input Mel-spectrograms concatenated with F0 and energy (RMS energy), obtained slightly lower average BLEU, ROUGE-1, ROUGE-2, and ROUGE-L scores of 62.8, but it achieved the highest disambiguation accuracy of 82.2%. This represents a significant improvement compared to the proposed cascade system.

## Limitations

In this work, the structurally ambiguous sentences created and handled by the disambiguation systems were adaptations of types 4, 5, 6, and 10 in the types of structurally ambiguous sentences in English by Taha (1983), adapted to Indonesian. Each structurally ambiguous sentence in this work was limited to having only two interpretations. Consequently, the sentences were relatively simple, with a maximum word count of 15. Furthermore, the utterances did not contain any disfluencies and were recorded in a low-noise environment. However, this study presents a potential solution for disambiguation. Therefore, further improvements will aim to address the current system limitations, such as allowing structurally ambiguous sentences to

have more than two interpretations and enabling the speech recognizer to handle disfluencies. Additionally, we will further enhance the performance of ASR, TD, and SD models to improve the disambiguation of structurally ambiguous sentences.

## Ethics Statement

This research study has been reviewed and approved by the Institutional Review Board of the Research Ethics Committee of the Japan Advanced Institute of Science and Technology (JAIST) under the research project titled "Construction of Multilingual AI Speech Language Technology That Imitates Human Communication to Overcome Language Barriers," in collaboration with Indonesian institutions (approval number 04-403, approval date: January 26th, 2023). Based on this approval, the experiments were conducted in accordance with the institutional ethics guidelines.

First, all annotators, speakers, and respondents in this study were recruited through public announcements. The selection process for crowdsourced participants followed a 'first in, first out' (FIFO) basis while ensuring that they met specific criteria: being native speakers of Indonesian, university students, and aged over 20 years old. These criteria were chosen based on the belief that university students who are native speakers would already possess strong Indonesian language skills.

Second, all participants were provided with information about research objectives, data usage and the protection of their personal information. We ensured that all annotators, speakers, and respondents involved in this study were fully aware that their data would be used exclusively for research purposes and would not be misused for other reasons. We removed any information that could identify individuals, such as speaker names, and anonymized them immediately after the experiment (e.g., speaker F01). Names were used only temporarily and not stored. It's important to note that while the speech data itself could potentially serve as a personal identification code, for this research, we needed to retain the speech data in its original form due to its nature.

Third, those participants who agreed provided written consent to participate in this study, in accordance with the institutional ethics guidelines. After the experiments, they received appropriate compensation based on the hourly work policy at our institution.

## Acknowledgements

Part of this work is supported by JSPS KAKENHI Grant Numbers JP21H05054 and JP21H03467, as well as JST Sakura Science Program.

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

# A  Hyperparameter Setting

The ASR and SD model training in this study did not involve hyperparameter tuning since they already employed almost identical hyperparameters to those used in the original Speech-transformer (Dong et al., 2018). The ASR and SD models were based on the Speech-transformer, which consisted of 12 encoder blocks, 6 decoder blocks, 4 heads, and a feed-forward inner dimension of 2048. We used 80 dimensions for the Mel-spectrogram input. The models were trained using the Adam optimizer (Kingma and Ba, 2014) with $\beta_1 = 0.9, \beta_2 = 0.98, \epsilon = 10^{-9}$ and employed cross-entropy loss with neighborhood smoothing. The learning rate varied during training with a warm-up period. Decoding was performed using beam search with a beam size of 10. The Speech-transformer model employed in this study is based on the ASR source code from the research conducted by Novitasari et al. (2022).

|          | Learning rate | Max epoch |
|----------|:-------------:|:---------:|
| Baseline | $10^{-4}$     | 10        |
| Cascade + tag | $10^{-4}$ | 50       |

Table 7: Hyperparameter tuning results for each TD model.

For TD, we aligned most hyperparameters with the optimal settings recommended for fine-tuning IndoBART, as provided by the IndoNLG paper (Cahyawijaya et al., 2021). These settings included a batch size of 8, early stopping after no improvements in 5 batches, a step decay learning rate scheduler with a step of 1 epoch and a gamma of 0.9, and fine-tuning with the Adam optimizer ($\beta_1 = 0.9, \beta_2 = 0.999, \epsilon = 10^{-8}$). However, we also conducted a simple hyperparameter tuning for the learning rate and maximum epochs within each tagging method (baseline and cascade+tag). We explored three combinations of learning rates and maximum epochs (learning rate=$10^{-4}$ and maximum epochs=10, learning rate=$10^{-4}$ and maximum epochs=50, learning rate=$10^{-5}$ and maximum epochs=10), while keeping the other hyperparameters consistent with the settings specified in the IndoNLG paper. The results of the hyperparameter tuning are presented in Table 7.