# OpenReview forum: "Speech Recognition and Meaning Interpretation: Towards Disambiguation of Structurally Ambiguous Spoken Utterances in Indonesian"
_EMNLP/2023/Conference — EMNLP 2023 Main_

### Official Review · Reviewer_iZHD · 2023-08-01

**Soundness:** 4

**Excitement:**

5: Transformative: This paper is likely to change its subfield or computational linguistics broadly. It should be considered for a best paper award. This paper changes the current understanding of some phenomenon, shows a widely held practice to be erroneous in someway, enables a promising direction of research for a (broad or narrow) topic, or creates an exciting new technique.

**Paper Topic And Main Contributions:**

Speech to text systems usually transcribe the exact utterance spoken into words. However, speech is often a lot more than what words can capture and this is what the authors exploit. They test whether adding information from speech helps in syntactic disambiguation. For this, they create a novel dataset in a low-resource Indonesian language -- speech -> tagged sentences -> disambiguated sentences.
They do this in two ways: (1) cascaded system, where the ASR, along with words, outputs some tags. Next, a text disambiguation block is used to finally disambiguate the sentence using these tags (2) end-to-end system that takes speech as the input and outputs disambiguated text. The authors report the performance in terms of 3 metrics: BLEU, ROUGE, and disambiguated text accuracy. The cascaded system performs better in terms of ROUGE and BLEU scores while the direct system performs better in terms of disambiguated text accuracy.

**Questions For The Authors:**

- [A] Why have the authors chosen end-to-end speech-former ASR models when there are only 40 hours (although seemed to have converged well looking at the results). I would have tried HMM-based ASRs or transfer learning + SSLs. Why was this model chosen over other good alternatives?
- [B] How would the system perform if the TD model had direct access to prosody information. In other words, in addition to text that has tags, if we input some prosody features extracted from speech to the TD model, would it do better?
- [C] No discussion or mention of the hyper parameters used in any of the setups (cascaded, direct). Is it all default hyperparameters? If so, could the authors add them in the appendix for easy reference. If not, could the authors show the results for all the hyperparameters they experimented with in the appendix?

**Reasons To Accept:**

- The paper is a delightful read (almost like an answer to Chrupała (2023)'s Putting "Natural" in Natural Language Processing, where he talks about going beyond just using speech to transcribe it to words)! A much needed new perspective on how speech can be used to solve difficult (using text-only methods) NLP tasks such as syntactic disambiguation.
- Amazing to also see this done in Indonesian! Usually such new problems statements are first explored in English/high resource languages and then applied to other low-resource languages.
- Nice comparison between cascaded systems and end-to-end systems
- A novel dataset that furthers research in this direction

**Reasons To Reject:**

- Done on read speech, which tends to be not so natural. But I can imagine conversational/spontaneous speech to be difficult to collect, especially for the task at hand.
- Problem of disambiguation seems to be oversimplified, which models having only 2 or 3 tags (<0>, <1>, <2>).

**Reproducibility:**

2: Would be hard pressed to reproduce the results. The contribution depends on data that are simply not available outside the author's institution or consortium; not enough details are provided.

**Reviewer Confidence:**

4: Quite sure. I tried to check the important points carefully. It's unlikely, though conceivable, that I missed something that should affect my ratings.

**Typos Grammar Style And Presentation Improvements:**

- Line 488/489 “with” and “without”
- Line 347/348 “which should have inherently preserves all..” doesn’t read well to me.
- BLEU and ROUGE is reported in terms of % in the abstract, which is not the norm.

---

> ### Author Rebuttal · Authors · 2023-08-29
>
> We express our sincere gratitude for the insightful and constructive review. Below we address the reviewer's concerns point by point.
> 1. **Done on read speech, which tends to be not so natural. But I can imagine conversational/spontaneous speech to be difficult to collect, especially for the task at hand**
>    Answer: In our work, we start with read speech to reduce variation and focus on ambiguity. Even so, we still maintain the naturalness of our corpus by informing the speakers to keep speaking naturally and monitoring each speaker during recording so that we can ask the speakers to correct their reading if their speech sounds unnatural.
>
> 2. **Why have the authors chosen end-to-end speech-former ASR models when there are only 40 hours (although seemed to have converged well looking at the results). I would have tried HMM-based ASRs or transfer learning + SSLs. Why was this model chosen over other good alternatives?**
>    Answer: Transformer has shown success in various natural language processing tasks, both in text and speech modalities. It can model long-range dependencies and be trained more effectively than recurrence sequence-to-sequence models. Therefore, we chose to use Transformer-based ASR to demonstrate that even with the limited data at our disposal, this powerful architecture can be used to train ASR effectively while still give adequate result.
>
> 3. **How would the system perform if the TD model had direct access to prosody information. In other words, in addition to text that has tags, if we input some prosody features extracted from speech to the TD model, would it do better?**
>    Answer: Incorporating prosodic features extraction into the TD model appears to be a good idea. However, in this work, we have not done that yet. We plan to do that in future research.
>
> 4. **No discussion or mention of the hyper parameters used in any of the setups (cascaded, direct). Is it all default hyperparameters? If so, could the authors add them in the appendix for easy reference. If not, could the authors show the results for all the hyperparameters they experimented with in the appendix?**
>    Answer: We will include the usage of hyperparameters if the paper is accepted.
>    * The ASR and SD models training in this study did not involve hyperparameter tuning since it already employed almost identical hyperparameters to the Speech-transformer paper.
>    * The IndoNLG paper provides the optimal hyperparameters for fine-tuning IndoBART. Nonetheless, we still attempted a simple hyperparameter tuning for the learning rate and maximum epochs within each tagging method (baseline and cascade+tag). We explored three combinations of learning rates and maximum epoches, while keeping the other hyperparameters consistent with the settings specified in the IndoNLG paper. The result of the hyperparameter tuning are presented in the table below.
>
>         |               | Best learning rate | Best max epoch |
>         |---------------|--------------------|----------------|
>         | Baseline      | $10^{-4}$    | 10             |
>         | Cascade + tag | $10^{-4}$    | 50             |
>
> We also appreciate your input on typos, grammar, style, and presentation improvements. If this paper is accepted, we will make the necessary enhancements.

---

### Official Review · Reviewer_PvXP · 2023-08-03

**Soundness:** 4

**Ethical Concerns:**

Yes

**Excitement:**

4: Strong: This paper deepens the understanding of some phenomenon or lowers the barriers to an existing research direction.

**Justification For Ethical Concerns:**

The paper makes use of numerous participants, at least some of which are crowdsourced. No information is given on their informed consent to participate or whether fair compensation was provided for their work. There is no ethics section which might clarify these points.

**Missing References:**

Lines 62 - you mention "most of the studies" it would be helpful to cite some examples

Line 115 - while you may know this from your own experience, there are a number of works that could be cited to demonstrate how Indonesian NLP lags behind. Consider whether a citation here might strengthen your statement.

**Paper Topic And Main Contributions:**

The paper addresses the problem of creating unambiguous interpretations in Indonesian transcribed speech, when then speech itself contains structural ambiguity (sentences which, as spoken/written, may have more than one meaning). To do this, they provide their model with prosodic information using specific signal processing techniques: root means square energy (RMSE) and mel-spectrogram to represent pauses; and Fundamental Frequency (F0) to represent pitch. To produce unambiguous text, three approaches are compared. The baseline approach uses ASR and a specially trained text disambiguation model. The proposed approaches use a modified ASR that attempts to tag one of three interpretations to the transcribed audio, then performs text disambiguation. The last approach is end-to-end. The paper demonstrates significant improvement with the proposed approaches over the baseline. The work also includes the preparation of a corpus for this task.

**Questions For The Authors:**

Question A: Will you be releasing the corpus you have prepared for this task?

Question B: Is sentence level the appropriate choice for evaluating structural ambiguity? For example, if the sentence "I saw someone on the mountain." is preceded with "I was about to take the trail leading up the mountain." then the context removes the ambiguity.

Question C: Can you describe, generally, the motivation for this research? If it is only to make the text less ambiguous, are there particular situations when you see that as more valuable? If you believe this may contribute to downstream tasks, which ones?

Question D: In line 369 you provided a survey to 20 participants for the human assessment. Did you ensure these participants are different than the crowdsourced participants mentioned earlier? Were these participants also crowdsourced?

Question E: Were the participants in your study compensated for their contribution in any way? Were they fully informed of how their contributions would be used?

**Reasons To Accept:**

The paper is well written in that it is clear to the approaches taken. It introduces a novel approach  and task in an under-served language for preserving the intent of speech in written form. The findings seem significant, even with a few methodological concerns.

**Reasons To Reject:**

No ethics section, but there are ethical issues that deserve discussion (see the ethics section).

Also a few, mostly minor points:

- When the corpus was created, participants were told to speak in such a way to make the intent of the speech unambiguous. This may lead to over-emphasis compared with natural speech. There was no mention of any evaluation of the data to avoid this.
- The corpus was created with only ambiguous sentences, and the non-ambiguous content was taken from another source. There is a chance that different recording qualities between news (LSCVR) and crowdsourced data could artificially raise the ability of the model to distinguish between ambiguous (tag 1 or 2) and non-ambiguous (tag 0) sentences.
- The amount of data used to train the text disambiguation model was significantly lower than the data used for training the end-to-end system. Given that the difference between the two proposed systems is only a few percentage points, it brings into question the conclusion that the direct model is clearly the better of the two (but they still are both demonstrably superior to the baseline).
- It would be hard to reproduce the fine tuning of the IndoBART model without a little more information. Was it fine-tuned for a certain number of steps, for example?

**Reproducibility:**

3: Could reproduce the results with some difficulty. The settings of parameters are underspecified or subjectively determined; the training/evaluation data are not widely available.

**Reviewer Confidence:**

4: Quite sure. I tried to check the important points carefully. It's unlikely, though conceivable, that I missed something that should affect my ratings.

**Typos Grammar Style And Presentation Improvements:**

line 054 - resulting should be results
line 073-075 - there seems to be a missing word in this sentence starting with "However"
line 230 - "how to build the system" seems vague and broad. I'd suggest what you are doing here is looking for specific systems that might help to preserve the prosodic information
Table 6 - second to last column in the "Mel" section is missing data.

---

> ### Author Rebuttal · Authors · 2023-08-29
>
> We express our sincere gratitude for the insightful and constructive review. Below we address the reviewer's concerns point by point.
> 1. **No ethics section, but there are ethical issues that deserve discussion. The paper makes use of numerous participants, at least some of which are crowdsourced. No information is given on their informed consent to participate or whether fair compensation was provided for their work. There is no ethics section which might clarify these points.**
>   Answer: Before the experiment, all speakers, validators, and subjective evaluators have provided written consent for participating in this study in accordance with the institutional ethics guidelines reviewed and approved by the Institutional Review Board of the Research Ethics Committee of our institution. We have also explained how their contributions would be used and provide compensation based on the hourly work policy at our institute.
>
> 2. **When the corpus was created, participants were told to speak in such a way to make the intent of the speech unambiguous. This may lead to over-emphasis compared with natural speech. There was no mention of any evaluation of the data to avoid this.**
>    Answer: To avoid over-emphasis, we have asked the speakers to keep speaking naturally. We also monitored each speaker during the recording and requested corrections if their speech sounded unnatural due to over-emphasis. Furthermore, before the recording session, we had provided the speakers with the structurally ambiguous texts that they would read, for them to study in advance.
>
> 3. **The corpus was created with only ambiguous sentences, and the non-ambiguous content was taken from another source. There is a chance that different recording qualities between news (LSCVR) and crowdsourced data could artificially raise the ability of the model to distinguish between ambiguous (tag 1 or 2) and non-ambiguous (tag 0) sentences.**
>    Answer: Even if non-ambiguous utterances are ignored, our system maintains a commendable disambiguation accuracy of 0.74 and 0.78 for cascade and direct systems respectively.
>
> 4. **The amount of data used to train the text disambiguation model was significantly lower than the data used for training the end-to-end system. Given that the difference between the two proposed systems is only a few percentage points, it brings into question the conclusion that the direct model is clearly the better of the two (but they still are both demonstrably superior to the baseline).**
>    Answer: The TD model is a pretrained IndoBART model which we finetuned with data described in Table 3. In addition, the ASR model and SD model were trained with the same data so it cannot be said that the cascade training data is less than the direct training data. However, we continued the training and found that the BLEU and ROUGE scores for the cascade system are differ slightly than the direct system, while the direct system's disambiguation accuracy is better than the cascade system with a difference of 2.5%. Thus, the direct system is just slightly better than the cascade system.
>
> 5. **It would be hard to reproduce the fine tuning of the IndoBART model without a little more information. Was it fine-tuned for a certain number of steps, for example?**
>    Answer: We will include the usage of hyperparameters if the paper is accepted.
>   The IndoNLG paper provides the optimal hyperparameters for fine-tuning IndoBART. Nonetheless, we still attempted a simple hyperparameter tuning for the learning rate and maximum epochs within each tagging method (baseline and cascade+tag). We explored three combinations of learning rates and maximum epoches, while keeping the other hyperparameters consistent with the settings specified in the IndoNLG paper. The result of the hyperparameter tuning are presented in the table below.
>     |               | Best learning rate | Best max epoch |
>     |---------------|--------------------|----------------|
>     | Baseline      | $10^{-4}$    | 10             |
>     | Cascade + tag | $10^{-4}$    | 50             |
>
>
> 6. **Will you be releasing the corpus you have prepared for this task?**
>    Answer: If the paper is accepted, we will release the corpus.
>
> 7. **Is sentence level the appropriate choice for evaluating structural ambiguity? For example, if the sentence "I saw someone on the mountain." is preceded with "I was about to take the trail leading up the mountain." then the context removes the ambiguity.**
>    Answer: While that idea might be valid, our study specifically concentrates on addressing structural ambiguity within a single sentence.
>
> 8. **Can you describe, generally, the motivation for this research? If it is only to make the text less ambiguous, are there particular situations when you see that as more valuable? If you believe this may contribute to downstream tasks, which ones?**
>    Answer: As can be seen from the results of our human evaluation, native speakers can handle structurally ambiguous utterances quite well. However, non-native speakers may find this task challenging. In fact, the existing speech translation systems, both speech-to-text and speech-to-speech, often overlook ambiguity during translation. Therefore, this structural ambiguity resolution system has the potential to be used in speech translation systems, both speech-to-text and speech-to-speech.
>
> 9. **In line 369 you provided a survey to 20 participants for the human assessment. Did you ensure these participants are different than the crowdsourced participants mentioned earlier? Were these participants also crowdsourced?**
>    Answer: Yes, all of the speakers and survey participants were crowdsourced, and there is no overlap between survey participants and speakers.
>
> 10. **Were the participants in your study compensated for their contribution in any way? Were they fully informed of how their contributions would be used?**
>    Answer: Yes. All speakers, validators, and subjective evaluators have been provided written consent for participating in this study in accordance with the institutional ethics guidelines reviewed and approved by the Institutional Review Board of the Research Ethics Committee of our institution. We have also explained how their contributions would be used and provide compensation based on the hourly work policy at our institute.
>
> We also appreciate your input on missing references, typos, grammar, style, and presentation improvements. If this paper is accepted, we will make the necessary enhancements.

---

### Official Review · Reviewer_S9Q7 · 2023-08-07

**Soundness:** 4

**Excitement:**

4: Strong: This paper deepens the understanding of some phenomenon or lowers the barriers to an existing research direction.

**Paper Topic And Main Contributions:**

This paper addresses structural ambiguity in Indonesian ASR through three key contributions:
(1) collecting an Indonesian speech corpus with 4800 utterances (800 pairs of ambiguous sentences covering four types of structural ambiguity and spoken by multiple speakers);
(2) human assessment study on how well speakers are able disambiguate structurally ambiguous sentences in both speech and text, giving context and an upper bound for later experiments;
(3) training ASR models and studying the impact of prosodic features (pitch and pause) as well as decoupling the ASR task from structural disambiguation (by either using a single direct model, or a cascade of ASR plus features/tagging and re-generation with a finetuned IndoBART model to generate unambiguous sentence(s). Experiments show that prosodic information is (expectedly) crucial for this task.

The paper is clearly written, with illustrative examples, and takes a thorough approach to the topic, from corpus creation, to speaker study, to model training and methodology. Some small table formatting cleanup would be helpful.
The inclusion of the human study is quite nice for context, showing the expected upper bound on text-only disambiguation and speech for downstream models, and that it differs slightly by type.
The created corpus is quite small, and read speech with subjects knowing the task which may lead to exaggerated or less natural prosody, but, would be challenging to collect otherwise and newly enables study of this particular phenomenon in Indonesian.

**Reasons To Accept:**

- Thorough treatment of a new area of study for Indonesian, including corpus creation, a human assessment, and experimental comparison

**Reasons To Reject:**

- Weaknesses are that the created corpus is quite small, not sure what differences may be significant;

**Reproducibility:**

3: Could reproduce the results with some difficulty. The settings of parameters are underspecified or subjectively determined; the training/evaluation data are not widely available.

**Reviewer Confidence:**

3: Pretty sure, but there's a chance I missed something. Although I have a good feel for this area in general, I did not carefully check the paper's details, e.g., the math, experimental design, or novelty.

**Typos Grammar Style And Presentation Improvements:**

The metrics (BLEU, ROUGE, WER, accuracy) in Tables 4-6 are typically reported in the 0-100 range; some scores appear to be formatted this way and some 0-1, please standardize (to 0-100).

Table 6: Please report only 1 decimal place for BLEU, ROUGE, or WER. The metrics are not sensitive enough to require 3 decimal places and it will make the table easier to read. Consider using \resizebox{} so ROUGE will not be broken onto 2 lines. Why is ROUGE-1 in the second to last columns in Table 6?  Perhaps state in the table caption what is being highlighted with red text

Additional training details to aid reproduction of experiments should be added at least to the appendix, for example number of finetuning steps, batch size, model vocabulary, etc.

Could you consider initializing the ASR decoder with IndoBART as an additional comparison to account for the additional training data the cascade model may have seen?

---

> ### Author Rebuttal · Authors · 2023-08-29
>
> We express our sincere gratitude for the insightful and constructive review. Below we address the reviewer's concerns point by point.
> 1. **Weaknesses are that the created corpus is quite small, not sure what differences may be significant**
>    Answer: One of the biggest problems encountered in research on the Indonesian language is the limited amount of data available, especially speech data. However, here we managed to show that even with small data there is a possibility for building an Indonesian speech disambiguation system.
>
> 2. **Could you consider initializing the ASR decoder with IndoBART as an additional comparison to account for the additional training data the cascade model may have seen?**
> Answer: Our focus in this research is to compare cascade and direct approaches of disambiguation systems with their ASR and SD using the same model. However, we will keep your suggestion in mind for future research possibilities.
>
> We also appreciate your input on typos, grammar, style, and presentation improvements. If this paper is accepted, we will make the necessary enhancements.

---

### Meta-Review · Area_Chair_j9ED · 2023-09-13

**Recommendation:** 5

**Metareview:**

While reviewers pointed out the size, naturalness of the dataset, simplification of the task, and minor ablation study, it seems the work verified itself with the importance of the task, well-designed experiment, original approach, and significant findings. The paper is well written and the evaluation metric is also reasonable to convince readers.

However, it still remains an ethical concern and would require additional evidence as suggested by ethics chairs.

---

### Decision · Program_Chairs · 2023-10-07

**Decision:**

Accept-Main

**Comment:**

While reviewers pointed out the size, naturalness of the dataset, simplification of the task, and minor ablation study, it seems the work verified itself with the importance of the task, well-designed experiment, original approach, and significant findings. The paper is well written and the evaluation metric is also reasonable to convince readers.

However, it still remains an ethical concern and would require additional evidence as suggested by ethics chairs.